# MYB Transcription Factors Becoming Mainstream in Plant Roots

**DOI:** 10.3390/ijms23169262

**Published:** 2022-08-17

**Authors:** Zhuo Chen, Zexuan Wu, Wenyu Dong, Shiying Liu, Lulu Tian, Jiana Li, Hai Du

**Affiliations:** 1College of Agronomy and Biotechnology, Chongqing Engineering Research Center for Rapeseed, Southwest University, Chongqing 400716, China; 2Academy of Agricultural Sciences, Southwest University, Chongqing 400716, China

**Keywords:** MYB transcription factors, plant roots, development, biotic and abiotic stresses

## Abstract

The function of the root system is crucial for plant survival, such as anchoring plants, absorbing nutrients and water from the soil, and adapting to stress. MYB transcription factors constitute one of the largest transcription factor families in plant genomes with structural and functional diversifications. Members of this superfamily in plant development and cell differentiation, specialized metabolism, and biotic and abiotic stress processes are widely recognized, but their roles in plant roots are still not well characterized. Recent advances in functional studies remind us that *MYB* genes may have potentially key roles in roots. In this review, the current knowledge about the functions of *MYB* genes in roots was summarized, including promoting cell differentiation, regulating cell division through cell cycle, response to biotic and abiotic stresses (e.g., drought, salt stress, nutrient stress, light, gravity, and fungi), and mediate phytohormone signals. *MYB* genes from the same subfamily tend to regulate similar biological processes in roots in redundant but precise ways. Given their increasing known functions and wide expression profiles in roots, *MYB* genes are proposed as key components of the gene regulatory networks associated with distinct biological processes in roots. Further functional studies of *MYB* genes will provide an important basis for root regulatory mechanisms, enabling a more inclusive green revolution and sustainable agriculture to face the constant changes in climate and environmental conditions.

## 1. Introduction

The plant root system provides the interface between plants and the complex soil environment, thereby being crucial for plant survival and crop productivity, such as absorbing nutrients and water from the soil, preventing lodging, and responding to biotic and abiotic stresses. In the past decades, plant root biology has attracted increasing attention as it has important implications for global food security under changing climate and environmental conditions [1,2,3,4,5]. The molecular basis of how the plant root system contributes to plant traits, such as lodging resistance, nutrient stress resistance, and biotic and abiotic stresses resistance are gradually clarified. It is commonly accepted that the root growth and development processes were regulated by complex mechanisms which include a series of transcription factors [6,7], such as *AtWOX5* controlling root apical meristem (RAM) division, *AtARF7/19* regulating the lateral roots (LRs) initiation, and *WEREWOLF* (*WER*/*AtMYB66*) determining the root hair (RH) cell formation [8,9,10].

The V-myb avian myeloblastosis viral oncogene homolog (MYB) transcription factors are commonly characterized by a highly conserved DNA-binding domain repeat (MYB domain) at the N-terminal, accompanied by a variable activation domain at the C-terminal [11,12]. MYB transcription factors (MYB TFs) are widely found throughout eukaryotic organisms and comprise a superfamily in land plant genomes, e.g., they account for ~13% of the 1500 transcription factors in the model plant *Arabidopsis thaliana* [13]. As compared with the large number in plant genomes, a few MYB TFs are generally present in unikonts, e.g., only three MYB TFs that are associated with cell cycle progression were reported in vertebrate animals including *Homo sapiens* and *Mus musculus* [14]; a single MYB TF that functions in cell cycle progression as well was present in invertebrates (such as *Drosophila melanogaster* and *Strongylocentrotus purpuratus*) [15]; similarly, only one MYB TF had been identified in cellular slime mold *Dictyostelium discoideum* [16]. In plants, based on the number of the highly conserved repeats (R) in the MYB domain, MYB proteins fall into four major families, namely MYB-related (R3/R1-MYB), 2R-MYB (R2R3-MYB), 3R-MYB (R1R2R3-MYB), and 4R-MYB (R1R2R2R1/R2-MYB) [17]. In general, the number of 3R-MYB and 4R-MYB families is very limited and conserved across plants, with most species possessing 3 and 2 members, respectively [17]. In contrast, numerous members are observed in MYB-related and 2R-MYB families in plants, especially in angiosperms [14,18]. Consistent with the small number, the functions of 3R-MYB and 4R-MYB families are conserved across plants and even eukaryotes. For example, the members of the 3R-MYB family in *Arabidopsis* play a similar role with their homologs in animals in cell cycle regulation [19,20,21]. On the contrary, the members of MYB-related and 2R-MYB families have diverse functions in many plant-specific processes [18,22]. To date, the major roles of *MYB*-related genes characterized are related to organ development, stress response, and circadian clock [18]. For example, CPC-like subfamily homologs are involved in cell fate determination [23], *AtMYBL* response to ABA and salt stresses [24], and *AtCCA1* with its homologs are involved in circadian rhythm [18]. Consistent with its vast number and phylogenetic classification, the 2R-MYB family has diverse functions in plants, which can be briefly summarized into three major biological processes: development and cell differentiation, biotic and abiotic stresses, and specialized metabolism [22]. For instance, the genes in the S15 subfamily are associated with cell fate determination [10], the genes in the S22 subfamily respond to drought and pathogen invasion [25], and those in the S6 subfamily participate in anthocyanin biosynthesis [26], etc. Due to inconvenient visualization, the root traits were generally neglected in plant genetics, biology, etc. Accordingly, to date, the functions of MYB proteins related to shoot attracted much more attention. However, an increasing number of *MYB* genes (*MYBs*) have been functionally characterized in roots, such as *At**MYB36* controlled LR primordium (LRP) development [27]; *At**MYB70* regulated root system development [28]; *At**MYB73*/*77* involved in LRs growth [29], etc. These results raised concerns about the roles of MYB TFs in roots. 

In this review, the functions of *MYBs* in plant roots are described in detail, with a focus on their roles in regulating cell differentiation, coordinating cell cycle, responding to biotic and abiotic stresses, and transducing phytohormone signals. The function characteristics of MYB homologs in roots have been discussed from a subfamily perspective. Finally, the potential research directions of MYB TFs in roots for future work were prospected. To our knowledge, this is the first review of the functions of MYB TFs in roots.

## 2. MYB TFs Are Regulators of Root Cell Differentiation

The development of plant roots relies on highly precise regulation of cellular differentiation [7]. MYB TFs were widely recognized to regulate the differentiation of root cells, participating in the development processes of primary root (PR), LRs, and RHs.

The roles of MYB TFs in root cell differentiation were represented by *WER* (2R-MYB) and *CAPRICE* (*CPC*, R3-MYB) which act as the core factors in determining root epidermal cell fate. In *Arabidopsis*, the specification of hair and non-hair epidermal cells are position-dependent, with hair cells arising over clefts between the overlying cortical cells (Figure 1A). In cortical cells, JACKDAW (JKD), a plant-specific zinc finger protein, produces a signal that may bind to the leucine-rich repeat receptor-like kinase (LRR-RLK) SCRAMBLED (SCM) in epidermal cells [30]. Due to the larger contact surface, the epidermal cell located at the cleft between two overlying cortical cells received more signals, resulting in more SCM activation [30]. Subsequently, the activated SCM represses the expression of the *WER* gene, contributing to more CPC than WER, leading to hair cell (HC) formation (Figure 1A) [31]. On the contrary, the epidermal cell on a single cortical cell has a little activated SCM because of the smaller contact surface, leading to the dominance of WER and thus developed as a non-hair cell (NHC) (Figure 1A) [31]. In NHC, a MYB protein WER, a bHLH protein GLABRA3 (GL3) or its homolog ENHANCER OF GLABRA3 (EGL3), and a WD40-repeat protein TRANSPARENT TESTA GLABRA1 (TTG1) form an activator protein complex MYB-bHLH-WD40 (MBW) to activate the expression of the downstream *GLABRA2* (*GL2*) gene that encodes a homeodomain protein [10,32,33,34]; subsequently, the GL2 protein represses a set of downstream genes such as *ROOT HAIR DEFECTIVE 6* (*RHD6*) and *RHD6-LIKE1* (*RSL1*) which are essential to the determination of RHs, leading to an NHC fate (Figure 1A) [35]. In HC, CPC protein competes with WER protein to form a repressor protein complex CPC-GL3/EGL3-TTG1 (MBW), which then inhibits the expression of the *GL2* gene [36,37]; thereby, the downstream genes of *GL2* (e.g., *RHD6* and *RSL1*) were normally expressed which then promote RH formation (Figure 1A).

The competition between the two types of MYB proteins, WER and CPC, to form the MBW ternary protein complex is attributed to their close evolutionary relationship. The MYB domain of CPC protein and the R3 repeat in the MYB domain of WER protein shared a high degree of sequence similarity. Both of them contain the conserved motif DLx2Rx3Lx6Lx3R in the R3 repeat of the MYB domains that are involved in the interactions between MYB and bHLH proteins [18]. However, as compared to WER, the CPC protein lacks the transcription activation domain causing its opposite function in the RH formation process [38]. Accordingly, the R3 domain of the WER can functionally replace CPC, but the R3 domain of CPC cannot functionally replace that of WER [39]. In addition, the homologs of *WER* and *CPC* genes were widely demonstrated to act as similar roles in RH formation process. For example, the *AtMYB23* gene has a similar function to the *WER* gene to reinforce the NHC fate [40]; in *Arabidopsis*, there are six homologs of the *CPC* gene, including *TRIPTYCHON*
*(TRY)*, *ENHANCER OF TRY AND CPC1* (*ETC1*), *ETC2*, *ETC3*, *TRICHOMELESS1* (*TCL1*)*,* and *TCL2*, which act as positive regulators for RH formation [23,41,42]. In other plant species, the homologs of *WER*/*CPC* genes were reported to perform similar functions in RH cell differentiation. For example, overexpressing four homologs of the *WER* gene in *Brassica napus* (*BnMYB019*, *BnMYB189*, *BnMYB231*, and *BnMYB388*) in *Arabidopsis* could rescue the RH phenotype of the *wer* mutant to that of WT lines [43]; expressing the *Solanum lycopersicum SlTRY* gene and *Oryza sativa* (rice) *OsTCL1* gene in *Arabidopsis*, respectively, both enhanced RH differentiation and thus promoted RH formation [44,45]. These results show a new perspective on subfamily to understand the key roles of *WER* and *CPC* homologs in root epidermal cell differentiation.

Notably, a series of feedback exists in this process to enhance root epidermal cell fate specification. In NHC, in addition to the downstream *GL2* gene, the WER-GL3/EGL3-TTG1 complex also positively regulates the expression of the competitors of the WER protein, *CPC* and *TRY*, and even the homologs of *WER*, *AtMYB23* (Figure 1A) [40,46]. However, the WER-GL3/EGL3-TTG1 complex negatively regulates the expression of *SCM*, *GL3*, and *EGL3* genes in NHCs as well [47]. At the same time, the GL3 and EGL3 proteins can move from HC to NHC through plasmodesmata (PD) [47], resulting in more WER-GL3/EGL3-TTG1 complex in NHC that can upregulate more GL2 and CPC/TRY protein formation (Figure 1A). Subsequently, the GL2 protein controls the NHC formation, whereas the CPC and TRY proteins can move from NHC to HC through PD to promote HC formation [37,48]. Moreover, TRY in HC can upregulate the expression of the *SCM* gene, contributing to the HC fate [49]. Recently, phytosulfokine receptors (PSKRs) which belong to the LRR-RLK family and an O-fucosyltransferase protein, SPINDLY (SPY), were reported to participate in position-dependent root epidermal cell fate determination as well [50,51]. Together, both the position signals and various feedback pathways determine the dominance of different MBW complexes in HCs and NHCs which consequently specify the root epidermal cell fate.

Another *2R-MYB* gene, *At**MYB3*6, is a key regulator of cell differentiation in *Arabidopsis* roots as well [27,52]. In endodermal cells, the *AtMYB36* gene is activated by SCARECROW (SCR) and then activates a series of genes such as *C**asparian strip proteins* (*CASPs*), *Peroxidase 64* (*PER64*) and *Enhanced Suberin 1* (*ESB1*), as well as represses *JKD* and *MAGPIE* (*MGP*) involved in proliferation, resulting in Casparian strip formation that represents a successful transition from proliferation to differentiation in endodermal cells (Figure 1C) [53]. Notably, the JKD-involved formation of the Casparian strip in the endodermal cells is independent of its function in epidermal patterning [47]. Moreover, *AtMYB36* plays a similar role in LRP, as the size and shape of LRP are determined by the balance between cell proliferation and differentiation as well (Figure 1B). In this biological process, *At**MYB36* is expressed in the cells surrounding LRP where it activates a subset of peroxidase genes such as *PER9* and *PER64* to regulate the ROS balance, consequently promoting the transition from proliferation to differentiation at later stages of LRP development [27]. Together, *AtMYB36* regulates the formation of the Casparian strip in endodermal cells and LRP development by promoting the transition from proliferation to differentiation in a similar manner.

Taken together, the roles of *MYBs* in cell differentiation in RH development were well-known in many plant species, whereas their roles in other root cell differentiation processes remain to be explored in the future.

## 3. MYB TFs Regulate Root Growth and Development through Cell Cycle

Root growth and development is a dynamic balance between cell proliferation and cellular differentiation, which rely on the positive and/or negative regulation of cell cycle progression. Another well-known role of plant MYB TFs in roots is controlling the cell cycle.

Among the MYB superfamily, the *3R-MYBs* were well-known to regulate the cell cycle in both animals and plants. In animals, the MYB family is composed of three members, *MYB* (*C-MYB*), *MYBL1* (*A-MYB*), and *MYBL2* (*B-MYB*), which play important roles in controlling the cell cycle. The *B-MYB* gene is ubiquitously expressed in all cell types and activates the transcription of *CDC2* and *cyclin B1* in the G2/M phase; the *A-MYB* and *C-MYB* exhibit a tissue-specific expression profile and are also involved in the regulation of cell cycle [54]. In *Arabidopsis*, four of the five *3R-MYBs* are demonstrated to regulate cell cycle progression, except *At**MYB3R2* which is associated with circadian rhythms rather than cell cycle [19,20,55]. Among them, *AtMYB3R1* and *AtMYB3R4* (*AtMYB3R1/4*) act as transcriptional activators which are expressed in proliferating tissues, such as root tips and LRPs (Figure 2A). In *atmyb3r1/4* double mutants, the expression of G2/M-specific genes was significantly reduced, leading to short roots and defective cytokinesis in root epidermal cells [20,56]. In contrast, *AtMYB3R1*, *AtMYB3R3*, and *AtMYB3R5* (*AtMYB3R1/3/5*) act redundantly as transcriptional repressors in both proliferating and differentiated cells by inhibiting the expression of G2/M-specific genes directly (Figure 2A). In *atmyb3r1/3/5* triple mutants, the expression of G2/M-specific genes was significantly upregulated, leading to an increased size of root meristems and a longer length of PR [19]. Commonly, the above *AtMYB3Rs* all bind to the *cis*-acting mitosis-specific activator (MSA) element (“AACGG”) in the promoter regions of several G2/M-specific genes (e.g., *AtKNOLLE*, *AtCYCB1,* and *AtCDC20*) to regulate the cell cycle (Figure 2A) [19,20]. However, the precise mechanism of how the *AtMYB3Rs* activate or repress the expression of downstream genes is still unclear. An explanation was that the AtMYB3Rs proteins can recruit different types of the DREAM (DP, RBR, E2F, and MuvB) complex which is well-known in the regulation of cell cycle [19]. In this process, the AtMYB3Rs indirectly recruit different E2F isoforms (E2FB or E2FC) through MuvB to form distinct DREAM complexes, such as AtMYB3R4-MuvB-E2FB and AtMYB3R3-MuvB-E2FC (Figure 2A,B) [57]. Consistently, E2FB functions as an activator while E2FC acts as a repressor in the regulation of cell cycle genes [58,59]. Notably, *AtMYB3R1* has dual roles in controlling the cell cycle, acting as an activator or a repressor in this process. In the *atmyb3r1* mutant, the expression of cell cycle genes underwent little change. However, enhanced downregulation and upregulation of G2/M-specific genes were observed when introducing *atmyb3r1* into *atmyb3r4* single mutant or *atmyb3r3/5* double mutants, respectively [19], demonstrating the dual roles of *AtMYB3R1* in the cell cycle. In RAM, the transcription of *AtMYB3R1* is downregulated by TSO1 (a core subunit of MuvB) rendering the cells unable to enter the differentiation process [60]. In other plants, *3R-MYB* homologs are demonstrated to perform a similar role in cell cycle progression. For instance, three *Nicotiana tabacum 3R-MYBs* (*NtmybA1*, *NtmybA2*, and *NtmybB*) and one rice *OsMYB3R-2* gene were reported to regulate the expression of cell cycle genes through binding to the MSA *cis*-element as well [61,62]. Recently, 225 *3R-MYBs* identified from 65 plant species appeared to be enriched for the MSA *cis*-element within their upstream promoter region, indicating a conserved functional involvement in cell cycle regulation [63]. Similarly, in animals, B-MYB is repressed by the DREAM complex in the G1/S phase while interacting with MuvB to activate the G2/M cell cycle genes [64,65]. These results suggest that *3R-MYBs* have undergone functional specialization during the evolution of plants.

The *2R-MYB**s* also regulate the cell cycle in roots, and their regulatory mechanisms appear to be spatiotemporally specific that is distinct from the broad regulation manner of *3R-MYBs*. For example, the *AtMYB59* gene is specifically expressed in roots and functions in the S phase of the cell cycle progression (Figure 2A) [66]. In *AtMYB59*-overexpressing lines, about half of the mitotic cells in root tips are at metaphase, leading to shorter PR than wild-type (WT); in contrast, the *atmyb59* mutants show the opposite phenotype, indicating that *AtMYB59* inhibits root growth by disturbing the cell cycle [66]. Different from *AtMYB3Rs*, *AtMYB59* upregulates the expression of cell cycle genes (such as *AtCYCB1;1*) by binding to the MRE (“AACC”), MRE2 (“TATAACGGTTTTTT”), and ERE (“ATTTCAAA”) *cis*-elements in the promoters instead of the MSA *cis*-element [66]. On the contrary, the *AtMYB56* (*BRAVO*) gene negatively regulates the expression of cell cycle genes (e.g., *AtCYCD2;2* and *AtCYCD3;3*) in quiescent center (QC) cells to ensure a low dividing activity (Figure 2B) [67]. Ectopic expression of *AtMYB56* resulted in the inhibition of root growth and the failure of root regeneration upon damage of stem cells [67]. Recently, the AtMYB56 protein has been reported to interact with the WOX5 protein to form a complex in QC cells (Figure 2B) [68]. In addition, an atypical 2R-MYB gene, *At**CDC5* (*cell division cycle 5*), is well-known for its role in regulating the cell cycle. This gene is predominantly expressed in proliferating cells and participates in the G2/M transition by upregulating the expression of the *AtCDKB1* gene (Figure 2A) [69]. Its binding site is the “CTCAGCG” motif in the promoters of target genes [69]. The root growth of the *AtCDC5*-RNAi plants was severely inhibited, and the *atcdc5* mutant was embryonic lethal, suggesting that *AtCDC5* is essential for cell cycle progression [69]. Moreover, these *2R-MYBs* commonly belong to the early derived subfamilies of the 2R-MYB family in plants, indicating that the ancient and important role of *2R-MYBs* in the cell cycle may be derived from *3R-MYBs*. To date, the knowledge regarding the roles of *2R-MYBs* in cell cycle progression is focused on the studies in *Arabidopsis*. However, as mentioned above, given the functional conservation characteristic of MYB homologs in diverse plant biological processes, it is foreseeable that *2R-MYBs* have an important role in root cell cycle progression in other plant species as well.

Overall, the regulatory roles of *3R-MYBs* in root cell cycle progression are widely recognized in many plants showing a conserved mechanism across different plant species, while the roles of *2R-MYBs* in root cell cycle are still poorly understood, especially in non-model plants.

## 4. MYB TFs Function in Root System Architecture in Response to Biotic and Abiotic Stresses

Due to the immobility, plants face diverse volatile environments during the life cycle, whereas the root system plays a key role and shows strong plasticity in response to abiotic stress. To date, a mass of studies has demonstrated that plant MYB TFs are involved in diverse abiotic stress responses, such as drought, salt stress, nutrient stress, gravity, and light (Table 1). Most of these processes were achieved by regulating root growth and/or development, root morphology, and root system architecture (RSA).

### 4.1. Drought

Drought is one of the most serious abiotic stresses affecting root growth. Under drought stress, the elongation of PR and the formation of LRs are adjusted to sustain the viability of plants. For example, as a typical adaptive response to environmental stress, PR growth can be promoted to absorb water deeper underground by sacrificing LRs development under drought conditions. MYB TFs have been reported frequently in drought resistance through modulating the RSA, including the length of PR and LRs as well as the number of LRs [25]. For example, in *Arabidopsis*, the *AtMYB96* gene was reported to negatively regulate LRs development and enhance drought resistance [81]; its overexpression lines showed significantly reduced LRs and enhanced drought resistance [81]. *AtMYB60*, a paralog of *AtMYB96*, promotes both PR and LRs growth to increase water uptake under mild drought stress [79]. In other plants, MYB TFs also regulate the RSA in response to drought. For instance, in *Glycine max*, *GmMYB84*-overexpressing lines exhibited enhanced drought resistance with a longer PR by controlling reactive oxygen species (ROS) balance [89]; in *Leymus chinensis*, *LcMYB2* improved plant drought resistance by promoting root growth as well [90]. However, MYB TFs play a negative role in drought response as well, e.g., overexpression of a wheat MYB gene *TaMpc1-D4* reduced the root length and repressed the expression of stress-related genes under drought stress [100].

### 4.2. Salt Stress

Salt stress is a major abiotic stress that adversely affects plant growth and development, significantly reducing crop productivity. Elevated soil salinity mainly causes ion toxicity and oxidative stress to roots, whereas MYB TFs are involved in these processes to respond to salt stress. *AtMYB30* enhances salt tolerance by improving alternative respiration which can maintain the root cellular redox homeostasis [73]. Moreover, *AtMYB30* links the reactive oxygen species (ROS) signaling and root cell elongation, and its mutant shows an increased cell length in root under H_2_O_2_ treatment [74]. Similarly, *AtMYB12* can upregulate the expression of ROS scavenging genes to maintain the root cellular ROS balance under both drought and salt stress conditions [71]. *AtMYB42* has been proven to participate in the regulation of ion toxicity. In this process, *AtMYB42* directly activates the expression of *salt overly sensitive 2* (*SOS2*), which plays a crucial role in regulating Na^+^:K^+^ homeostasis, resulting in the root system avoiding ion toxicity [75]. *AtMYB20* enhances salt tolerance by downregulating the expression of *type 2C serine/threonine protein phosphatases* (*PP2Cs*) that plays a negative role in ABA signaling [72]. The *AtMYB20*-overexpressing lines exhibit enhanced salt tolerance with a longer PR [72]. These results suggest that MYB TFs are involved in the regulation of diverse physiological processes to alleviate and even avoid the effects of salt stress, thereby ensuring root growth and development.

### 4.3. Nutrient Stress

Soil nutrient limitations, such as phosphorus (P), nitrogen (N), and potassium (K) starvation, are major abiotic stresses that affect plant growth and development and crop production. Consequently, roots have evolved a set of mechanisms to enhance nutrient acquisition, including changing RSA to enlarge root surface area to uptake nutrients from soil. Recently, an increasing number of *MYBs* has been reported to be involved in nutrient stress response processes by regulating the RSA, especially in response to P starvation. In *Arabidopsis*, a MYB-like gene *P Starvation Response 1* (*PHR1*), and its homologs *AtPHR1-like 1* (*AtPHL1*) and *AtPHL2* are recognized as central regulators of P starvation response (PSR) by directly regulating various *P starvation-induced* (*PSI*) genes, consequently affecting the P uptake and transport as well as modulating the RSA [83,84,85,102]. In *Oryza sativa*, four orthologs of *AtPHR1* (*OsPHR1/2/3/4*) function redundantly in a highly conserved manner to that of *AtPHR1* [95,96]. Two target genes of *AtPHR1*, *AtHRS1*, and *AtHHO2* (*HRS1 Homolog 2*) are MYB-like genes that participate in the regulation of RSA to enhance the adaptability to P starvation as well [103]. Under P starvation, *AtHRS1*-overexpression lines exhibited enhanced RHC development and shortened PR length, while *AtHHO2*-overexpression lines showed augmented LRs development [86,87]. Moreover, several *R3-MYBs*, *CPC*, *TRY*, and *ETC1* were well-known to play key roles in P starvation response by positively regulating RH development [88].

*2R-MYBs* participate in PSR as well. Under P starvation, *AtMYB2*-overexpression lines exhibited more and longer LRs and denser RHs [70]; the overexpression of *OsMYB2P-1* and *OsMYB4P* both enhanced the P starvation tolerance with enhanced root growth [93,94]. These three genes act as a positive regulator in PSR by activating the expression of downstream *PSI* genes, such as *miR399* and *P transporter* (*PHT*) genes. In contrast, *AtMYB62* acts as a negative regulator of the PSR by repressing the expression of *PSI* genes, such as *AtPHT1* and *AtACP5* [80]. Overexpression of the *AtMYB62* gene significantly decreased LRs length [80]. Rice 2R-MYB gene, *OsMYB1*, coordinately regulates the maintenance of P homeostasis and root development under P starvation conditions [92]. Moreover, MYB TFs were involved in adventitious root (AR) formation under P starvation, e.g., in *Populus ussuriensis*, *PuMYB40* controlled the P starvation resistance by promoting AR formation [98]. In addition, MYB TFs also participate in responding to other nutrient stress processes through modulating the RSA. Recently, *AtMYB59* has been reported to modulate the RSA in response to low K^+^, low NO_3_^−^, or low calcium (Ca) stresses [76,77], suggesting its multiple roles in nutrient stress responses.

### 4.4. Light and Gravity

Light is an important signal that regulates root growth and development, especially ultraviolet B (UV-B) light which is a part of sunlight that markedly affects root morphology [104]. Recently, *AtMYB73* and *AtMYB77* genes were reported to directly interact with the UV-B photoreceptor UVR8 (UV Resistance Locus 8) to regulate root growth under UV-B light in *Arabidopsis* [29]. In this process, AtMYB73 and AtMYB77 proteins interact with auxin response factors to promote LRs growth and development; however, UVR8 can interact with AtMYB73/77 proteins in a UV-B-dependent manner to inhibit their DNA-binding activities, consequently inhibiting the LRs development [29]. The root system can change the growth direction in response to gravity stimulation, and *MYBs* are involved in this process. It was reported that *AtMYB88* and *AtMYB124 (AtFLP)* performed a redundant role in response to gravity stimulation by regulating the temporal–spatial expression patterns of *PIN-FORMED 3* (*PIN3*) and *PIN7* genes in gravity-sensing cells of primary and lateral roots [82]. After gravity stimulation (reorientation of 90°), the curvature of PR in the *atflp* mutant exhibited a defective gravity response. Moreover, *AtMYB88* and *AtFLP* functioned complementarily in establishing the gravitropic set-point angles, with the former functions in later stages while the latter in earlier stages [82]. Similarly, in *Malus×domestica*, *MdFLP*, an ortholog of *AtFLP*, regulated ARs gravitropism by directly binding to the promoters of *MdPIN3* and *MdPIN10* genes [91].

### 4.5. Biotic Stress

To date, many *MYBs* had been demonstrated to have an important role in biotic stress resistance in aerial parts, e.g., *AtMYB30* and *AtMYB72* participated in the response to pathogen attack in *Arabidopsis* leaf [105,106]; *AtMYB15*, *AtMYB34*, *AtMYB51*, and *AtMYB75* were involved in the resistance against insect herbivores [107]. In recent years, a few studies revealed the roles of *MYBs* in response to biotic stress in roots as well, such as pest and disease stress responses. In wheat, the *TaPIMP2* gene plays a positive role in defense responses to fungal pathogen *Bipolaris sorokiniana* infection by regulating the expression of defense-related genes, contributing to the host resistance to common root rot [101]. In *Panax notoginseng*, the *PnMYB2* gene has a positive role in root rot resistance caused by *Fusarium solani* pathogen through regulating JA signaling, disease-resistance-related genes, and photosynthesis [97]. The *Thinopyrum intermedium TiMYB2R-1* gene exhibits enhanced resistance to the take-all disease in wheat by upregulating the expression of defense-related genes [99]. In *Arabidopsis*, the *AtMYB59* gene was found to play an important role in the response to *Heterodera schachtii* infestation in roots [78]. Recently, based on transcriptome and co-expression network analyses in *Brassica rapa*, several *MYBs* were speculated to be involved in clubroot resistance caused by the soil-borne protist *Plasmodiophora brassicae* [108]. These studies demonstrated that *MYBs* are also involved in biotic stress resistance in roots, which is an important direction for future gene functional research on this gene family in plants.

## 5. The Regulatory Roles of MYB in Roots Are Mediated by Various Phytohormones

Phytohormones are involved in many biological processes in roots, such as root growth and development, root morphogenesis, stress response, and signal transduction. In fact, it was demonstrated that the roles of MYB TFs in roots are widely mediated by various phytohormones, including indole acetic acid (IAA), abscisic acid (ABA), gibberellin (GA), cytokinin (CTK), and brassinosteroid (BR). Meanwhile, MYB TFs are also involved in phytohormone response processes to regulate root growth and development and stress response by activating downstream phytohormone-related genes and/or acting as response factors to regulate various physiological biochemical processes in roots (Figure 3).

The IAA-mediated MYB signaling transduction pathways are widely involved in the regulation of diverse developmental and stress response processes in roots. In most cases, *MYBs* directly regulate the expression of IAA-related genes to modulate the RSA. For instance, in *Arabidopsis*, *AtMYB88* and *AtMYB124* participate in root gravitropism by regulating the transcription of *PIN3* and *PIN7* genes that encode auxin transporter proteins [82]; *AtMYB96* upregulates the expression of the *GH3* gene which encodes the auxin-conjugating enzyme to negatively regulate LRs growth [81]. In turn, IAA could induce the expression of *MYBs* to perform their biological function in roots. For example, the expression of *AtMYB96* could be induced by IAA leading to reduced LRs and enhanced drought resistance in *Arabidopsis* [81]; *AtMYB93* is a novel auxin-induced negative regulator in LRs development [109].

ABA is well-known for its key role in abiotic stress. Many MYB TFs are widely involved in the ABA signaling pathway to modulate RSA. For example, *AtMYB30* regulates root cell elongation depending on the ABA signaling [110]; *AtMYB20* modulates the RSA by downregulating the expression of *ABI1* and *PP2C* genes which encode the key regulators of the ABA signaling pathway [72]; *AtMYB12* modulates the RSA under salt and drought stresses by upregulating the ABA biosynthesis genes (e.g., *AtZEP*, *AtNCED*, *AtABA2*, and *AtAAO*) [71]. In most cases, the roles of *MYBs* in roots were addressed by integrating signaling of different phytohormones, such as ABA-IAA. For instance, *AtMYB70* regulates the RSA through activating the expression of the *GH3* gene under ABA signaling [28]; *AtMYB44/73/77* promotes LRs growth by activating the auxin-responsive genes under ABA signaling [111]; *AtMYB96* is a molecular link that integrates ABA and IAA signaling in LRs growth under drought stress [81].

*MYBs* can regulate root development through other phytohormone signal pathways, such as GA, CTK, and BR. *AtMYB62* and *OsMYB1* function in the modulation of RSA under P starvation by changing the GA metabolism and signaling [80,92]. *AtMYB11*, *AtMYB12*, and *AtMYB111* positively regulate root growth under the presence of GA signaling by affecting the IAA content, suggesting that *AtMYB11/12/111*-mediated GA signals are integrated into the IAA signaling pathway [112]. In addition, CTK promotes the nuclear localization of AtMYB3R4 protein which is essential for cell division in both SAM and RAM [113]; *AtMYB56* acts as a switch to modulate the QC cells division by interacting with the BR-regulated transcription factor BRI1-EMS SUPPRESSOR 1 (BES1) [67]. Thus, the roles of MYB TFs in roots were generally mediated by phytohormone-mediated signaling pathways.

## 6. Conclusions and Prospects

The root system plays an essential role in plant anchorage, water, nutrient acquisition, and stress response. Therefore, genetic improvement of the root system has been recognized gradually as an effective strategy, which contributes to improving crop productivity under diverse growing conditions and thus enables a more inclusive green revolution. In the past decade, an increasing number of *MYB**s* have been experimentally demonstrated to play critical roles in nearly all areas of root systems, such as PR, LRs, RHs, and ARs growth and development, and responding to diverse biotic and abiotic stresses. Among the MYB superfamily, *3R-MYBs* maintain their cell cycle regulatory roles in roots, whereas *1R*-*MYBs* and *2R-MYBs* function in root regulation in various ways, including cell cycle, cell differentiation, biotic and abiotic stresses, and phytohormone signaling pathways. The current understanding shows the fact that MYB TFs are becoming mainstream in plant roots, showing important application potential in molecular and genetic manipulation of root traits.

The MYB gene family is one of the largest transcription factor families in plant genome, with hundreds of members in most angiosperms. However, compared with its larger number, relatively limited members have been functionally characterized to date. The current knowledge regarding the functions of plant *MYBs* is mainly based on the studies in model plant *Arabidopsis*, these in non-model plants, especially in crops, remain rather unclear. Fortunately, an increasing number of sequenced genome and transcriptome data are available, which enabled genome-wide identification, systematic analysis, and overview of this gene family in diverse plant species. Consequently, the MYB superfamily has been widely identified and functionally verified in many plants, such as maize, soybean, and rapeseed. Based on the genome-wide expression analyses, a large proportion of *MYB*s have been proved to be preferentially and even specifically expressed in roots. Undoubtedly, much wider roles of *MYBs* in plant roots will be continuously recognized in the near future based on the efforts from both of the traditional and modern disciplines, such as Molecular Biology, Genetics, Cell Biology, Genomics, multi-omics, etc.

## Figures and Tables

**Figure 1 ijms-23-09262-f001:**
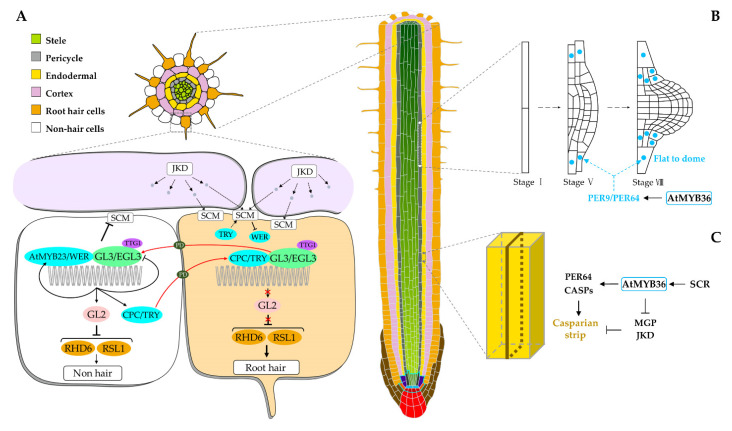
MYB transcription factors (MYB TFs) promote cell differentiation in *Arabidopsis* roots. (**A**) The model of *MYB* genes (*MYBs*) controls root epidermal cell specification. In non-hair cell (NHC), the WER-GL3/EGL3-TTG1 protein complex inhibits root hair (RH) formation by activating the *GL2* gene expression which then inhibits the expression of the downstream genes such as *RHD6* and *RSL1* to repress RH formation. In hair cell (HC), the CPC-GL3/EGL3-TTG1 complex cannot active *GL2* gene expression in HCs, thereby the downstream genes (e.g., *RHD6* and *RSL1*) can promote RH formation. The dominance of different MBW complexes in HCs and NHCs is determined by the position signals (grey dots) in NHCs and HCs and the lateral movements between NHCs and HCs (red arrows). Blue dots represent MYB proteins. (**B**) The model of the *AtMYB36* gene regulates lateral root development. Blue dots represent the expression position of the *AtMYB36* gene. (**C**) The model of *AtMYB36* regulates Casparian strip formation. *AtMYB36* promotes the formation of the Casparian strip by regulating the balance between proliferation and differentiation. Brown lines indicate the Casparian strip.

**Figure 2 ijms-23-09262-f002:**
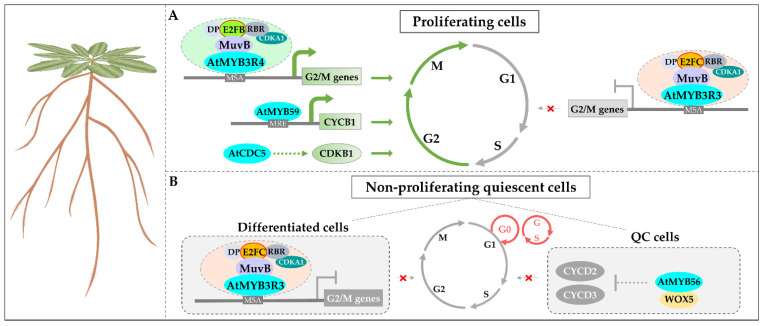
MYB TFs regulate the cell cycle in *Arabidopsis* roots. (**A**) In proliferating cells, AtMYB3R4 activates the expression of G2/M-specific genes by interacting with the DREAM complex containing E2FB at the G2/M phase, while AtMYB3R3 interacts with the DREAM complex containing E2FC to inhibit G2/M-specific gene expression at the G1/S phase. AtMYB59 activates *CYCB1* to promote G2/M transition in the root apical meristem (RAM). AtCDC5 participates in cell cycle progression by positively regulating *CDKB1* expression at the G2/M phase in RAM. (**B**) In non-proliferating quiescent cells, AtMYB3R3 interacts with the DREAM complex containing E2FC to inhibit the expression of G2/M-specific genes to maintain the quiescent state in differentiated cells; AtMYB56 represses the expression of *CYCD2* and *CYCD3* genes by interacting with WOX5 to maintain a low division rate of quiescent center cells. The blue dots represent MYB proteins.

**Figure 3 ijms-23-09262-f003:**
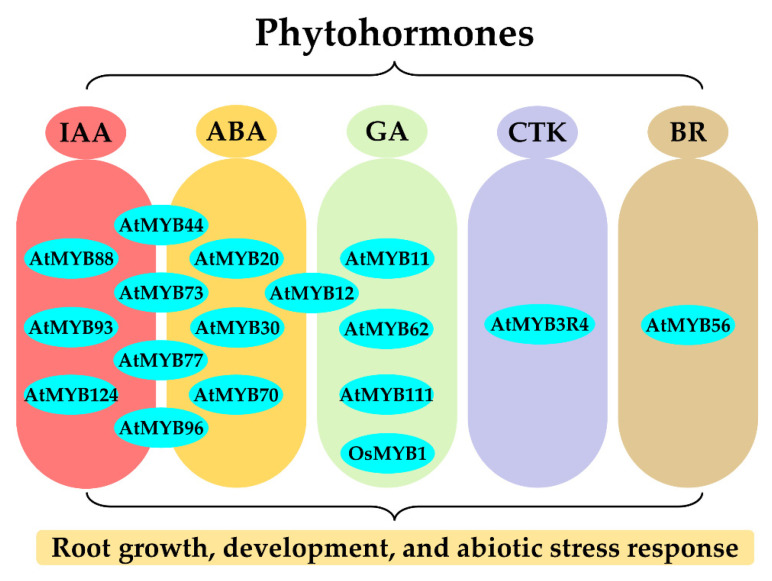
The regulatory roles of *MYBs* in roots are mediated by various phytohormones. IAA: indole acetic acid; ABA: abscisic acid; GA: gibberellin; CTK: cytokinin; BR: brassinosteroid. The blue dots represent MYB TFs. The *MYBs* crosslink different types of phytohormones indicating they can integrate different phytohormone signaling.

**Table 1 ijms-23-09262-t001:** MYB gene functions in roots under biotic and abiotic stresses.

Species	Genes	Function Description	References
*Arabidopsis thaliana*	*AtMYB2*	Positively regulate LRs and RHs under P starvation	[70]
	*AtMYB12*	Maintain root growth under salt and drought stress	[71]
	*AtMYB20*	Promote PR growth under salt stress	[72]
	*AtMYB30*	Regulate root growth and development under salt stress	[73,74]
	*AtMYB42*	Protect root from ion toxicity under salt stress	[75]
	*AtMYB59*	Modulate RSA under nutrient stress and participate in the response to nematode infestation	[76,77,78]
	*AtMYB60*	Promote root growth under mild drought	[79]
	*AtMYB62*	Negative regulators of LRs growth under P starvation	[80]
	*AtMYB73/77*	Mediate the inhibition of LRs under UV-B light	[29]
	*AtMYB96*	Negatively regulate LRs under drought	[81]
	*AtMYB88/124*	Participate in PR and LRs gravitropism	[82]
	*AtPHR1*	A central regulator of P starvation	[83]
	*AtPHL1/2*	Dimerize with AtPHR1 to regulate P starvation responses	[84,85]
	*AtHRS1*	Regulate RHs and PR under P starvation	[86]
	*AtHHO2*	Promote LRs under P starvation	[87]
	*CPC/TRY/ETC1*	Positively regulate RHs development under P starvation	[88]
*Glycine max*	*GmMYB84*	Promote PR elongation under drought	[89]
*Leymus chinensis*	*LcMYB2*	Positively regulate root growth under drought	[90]
*Malus*×*domestica*	*MdFLP*	Promotes ARs in response to gravity	[91]
*Oryza sativa*	*OsMYB1*	Regulate LRs elongation under P starvation	[92]
	*OsMYB2P-1*	Positively regulate root growth under P starvation	[93]
	*OsMYB4P*	Promote PR growth under P starvation	[94]
	*OsPHR1/2/3/4*	A central regulator of P starvation	[95,96]
*Panax notoginseng*	*PnMYB2*	Regulate the resistance against the root rot	[97]
*Populus ussuriensis*	*PuMYB40*	Promote ARs formation under P starvation	[98]
*Thinopyrum intermedium*	*TiMYB2R-1*	Enhance the resistance to take-all disease	[99]
*Triticum aestivum*	*TaMpc1-D4*	Negative regulators with reduced root growth under drought	[100]
	*TaPIMP2*	Contribute to wheat resistance to root rot	[101]

PR, primary root; LRs, lateral roots; ARs, adventitious roots; RHs, root hairs; RSA, root system architecture; UV-B, ultraviolet B; P, phosphorus.

## Data Availability

The study did not report any data.

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
