# Peer review of "MYB Transcription Factors Becoming Mainstream in Plant Roots"

_ijms, 2022, doi:10.3390/ijms23169262_

Round 1

Reviewer 1 Report

The current study reports an interesting topic that points out the MYB transcription factors becoming mainstream in plant roots. The manuscript shows high originality and novelty and minor adjustments needed in standard English. The presented parts are significant and interpreted appropriately. The raised conclusions and further suggestions are justified. The study covers its topic which is well relevant and all used references are appropriate. Also, the study is correctly designed and sounds technically.

The Abstract part is well written, aiming and clear. Just the first voice form of the sentences should be avoided and the impersonal form should be always adopted instead. All keywords fit well. The Introduction part is well structured and aiming. It needs minor adjustments in terms of linguistic mistakes and sentences reformulation in the impersonal form rather than the first voice’s one. However, the study’s aims are very clear and interesting. The “MYB TFs are regulators of root cells differentiation”, “MYB TFs regulate root growth and development through cell cycle” and “The regulatory roles of MYB in roots are mediated by various phytohormones” parts are well structured and clear. The “MYB regulators involved in root system architecture in response to abiotic stress” part is aiming and well structured but needs minor adjustments related to linguistic mistakes. The Conclusions and Prospects part is very well structured and aiming. However, I suggest removing all in-text references found in this part as this is not appropriate.

Briefly, based on the above and below detailed explanation, the manuscript needs minor adjustments, but have a big merit to be published in “International Journal of Molecular Sciences” (IJMS) once all suggestions and recommendations are fully addressed.

Abstract

1)      Page 1, lines 10–25: The Abstract part is well written, aiming and clear. Just kindly avoid the first voice form of the sentences and adopt always the impersonal form instead. On the other hand, all keywords fit well.

2)      Page 1, lines 16–22: “In this… roots”: Kindly avoid the first voice form of these sentences and adopt the impersonal form instead.

3)      Page 1, line 26: All keywords fit well.

1. Introduction

1)      Pages 1–2, lines 29–77: The Introduction part is well structured and aiming. It needs minor adjustments in terms of linguistic mistakes and sentences reformulation in the impersonal form rather than the first voice’s one. However, the study’s aims are very clear and interesting.

2)      Page 1, line 42: Kindly adjust as follow: “account”.

3)      Page 2, line 69: Kindly adjust as follow: “controlled”.

4)      Page 2, line 70: Kindly adjust as follow: “regulated”.

5)      Page 2, line 71: Kindly remove “our”.

6)      Page 2, lines 72–76: “In this… work”: Kindly avoid the first voice form of these sentences and adopt the impersonal form instead.

2. MYB TFs are regulators of root cells differentiation

1)      Pages 2–4, lines 79–167: The “MYB TFs are regulators of root cells differentiation” part is well structured and clear.

2)      Page 3, line 125: Kindly adjust as follow: “as compared to”.

3. MYB TFs regulate root growth and development through cell cycle

1)      Pages 4–6, lines 169–245: The “MYB TFs regulate root growth and development through cell cycle” part is clear and well aiming.

4. MYB regulators involved in root system architecture in response to abiotic stress

1)      Pages 6–9, lines 247–343: The “MYB regulators involved in root system architecture in response to abiotic stress” part is aiming and well structured. It needs minor adjustments related to linguistic mistakes.

2)     Page 6, line 249: Kindly remove “had”.

3)     4.1. Drought: Page 7, line 270: Kindly adjust as follow: “improved”.

4)     4.2. Salt stress: Page 8, line 281: Kindly adjust as follow: “shows”.

5)     4.2. Salt stress: Page 8, line 289: Kindly adjust as follow: “exhibit”.

6)   4.3. Nutrient stress: Page 8, line 294: Kindly adjust as follow: “phosphorus (P)”. Kindly pay attention to adjust this symbol in the whole manuscript.

7)      4.3. Nutrient stress: Page 8, line 298: Kindly adjust as follow: “has been reported”.

8)      4.3. Nutrient stress: Page 8, line 314: Kindly adjust as follow: “enhanced”.

9)      4.3. Nutrient stress: Page 8, line 322: Kindly adjust as follow: “controlled”.

10)  4.4. Light and gravity: Page 9, line 342: Kindly adjust as follow: “regulated”.

5. The regulatory roles of MYB in roots are mediated by various phytohormones

 1)      Pages 9–10, lines 345–392: “The regulatory roles of MYB in roots are mediated by various phytohormones” part is well structured and aiming.

6. Conclusions and Prospects

 1)      Pages 10–11, lines 394–421: The Conclusions and Prospects part is very well structured and aiming. However, I suggest removing all in-text references found in this part as this is not appropriate.

2)      Page 10, line 409: Kindly remove “And” from the beginning of the sentence.

Author Response

Dear Reviewer#1,

Thank you very much for your kindly suggestions in our manuscript “MYB transcription factors becoming mainstream in plant roots” (Manuscript ID: ijms-1860833). We have revised the manuscript according to your comments, carefully. In the following pages are our point-by-point responses to each comment of you. According to the journal’s requirement, all of the revisions in the manuscript were marked up using the “Track Changes” function in the Microsoft word. Moreover, we use red highlight for the main amends in the manuscript. In addition, according to Reviewer #2’s suggestion, we have added some new contents about the functions of MYB genes involved in roots in none-model plants, and have also added a new part about the functions of MYB genes in response to biotic stress in roots in the revised manuscript (Page 10). As a result, the number of references in the main text and the “Reference” parts were adjusted accordingly. Please check them. Thank you. We hope that our revisions in the manuscript and our accompanying responses will be sufficient to make our manuscript suitable for publication in your journal.

Thank you very much for your hard work in reviewing our manuscript.

Yours sincerely,

Hai Du

Reviewer#1

 Comments and Suggestions for Authors:

The current study reports an interesting topic that points out the MYB transcription factors becoming mainstream in plant roots. The manuscript shows high originality and novelty and minor adjustments needed in standard English. The presented parts are significant and interpreted appropriately. The raised conclusions and further suggestions are justified. The study covers its topic which is well relevant and all used references are appropriate. Also, the study is correctly designed and sounds technically.

The Abstract part is well written, aiming and clear. Just the first voice form of the sentences should be avoided and the impersonal form should be always adopted instead. All keywords fit well. The Introduction part is well structured and aiming. It needs minor adjustments in terms of linguistic mistakes and sentences reformulation in the impersonal form rather than the first voice’s one. However, the study’s aims are very clear and interesting. The “MYB TFs are regulators of root cells differentiation”, “MYB TFs regulate root growth and development through cell cycle” and “The regulatory roles of MYB in roots are mediated by various phytohormones” parts are well structured and clear. The “MYB regulators involved in root system architecture in response to abiotic stress” part is aiming and well structured but needs minor adjustments related to linguistic mistakes. The Conclusions and Prospects part is very well structured and aiming. However, I suggest removing all in-text references found in this part as this is not appropriate.

Briefly, based on the above and below detailed explanation, the manuscript needs minor adjustments, but have a big merit to be published in “International Journal of Molecular Sciences” (IJMS) once all suggestions and recommendations are fully addressed.

Abstract

1) Page 1, lines 10–25: The Abstract part is well written, aiming and clear. Just kindly avoid the first voice form of the sentences and adopt always the impersonal form instead. On the other hand, all keywords fit well.

Response: Thank you for your kindly suggestion. We have amended this issue in the “Abstract” part. Thanks you.

2) Page 1, lines 16–22: “In this… roots”: Kindly avoid the first voice form of these sentences and adopt the impersonal form instead.

Response: We have already amended it to avoid the first voice form of the sentences in our manuscript. Thank you for your comment.

3) Page 1, line 26: All keywords fit well.

Response: Thank you for your positive comment about the “Keywords”. In addition, according to Reviewer#2’s requirement, we adjusted the “abiotic stress” to “biotic and abiotic stresses” in this part. Thank you.

  1. Introduction

1) Pages 1–2, lines 29–77: The Introduction part is well structured and aiming. It needs minor adjustments in terms of linguistic mistakes and sentences reformulation in the impersonal form rather than the first voice’s one. However, the study’s aims are very clear and interesting.

Response: Thank you for your comments. We have corrected this mistake in the “Introduction” part. Moreover, we have checked this issue in the full text to avoid the same problem. Thank you again.

2) Page 1, line 42: Kindly adjust as follow: “account”.

Response: We have corrected this problem in our manuscript, according to your suggestion. Thank you.

3) Page 2, line 69: Kindly adjust as follow: “controlled”.

Response: We have adjusted “controls” into “controlled” in our manuscript. Thank you for your suggestion.

4) Page 2, line 70: Kindly adjust as follow: “regulated”.

Response: We have adjusted “regulates” to “regulated” in our manuscript. Thank you for your suggestion.

5) Page 2, line 71: Kindly remove “our”.

Response: We have already removed it in our manuscript. Thank you.

6) Page 2, lines 72–76: “In this… work”: Kindly avoid the first voice form of these sentences and adopt the impersonal form instead.

Response: Thank you for your comment. We have amended this problem in our manuscript carefully.

  1. MYB TFs are regulators of root cells differentiation

1) Pages 2–4, lines 79–167: The “MYB TFs are regulators of root cells differentiation” part is well structured and clear.

Response: We really appreciate your positive comments about this part. Thank you.

2) Page 3, line 125: Kindly adjust as follow: “as compared to”.

Response: We have adjusted this issue in our manuscript. Thank you for your suggestion.

  1. MYB TFs regulate root growth and development through cell cycle

1) Pages 4–6, lines 169–245: The “MYB TFs regulate root growth and development through cell cycle” part is clear and well aiming.

Response: Thank you very much for your positive comments about this part.

  1. MYB regulators involved in root system architecture in response to abiotic stress

1) Pages 6–9, lines 247–343: The “MYB regulators involved in root system architecture in response to abiotic stress” part is aiming and well structured. It needs minor adjustments related to linguistic mistakes.

Response: We are sorry for these mistakes. We have corrected them in our revised manuscript according to your suggestion. Thank you.

2) Page 6, line 249: Kindly remove “had”.

Response: We have removed it in our manuscript. Thank you.

3) 4.1. Drought: Page 7, line 270: Kindly adjust as follow: “improved”.

Response: We have adjusted “improves” into “improved” in our manuscript. Thank you.

4) 4.2. Salt stress: Page 8, line 281: Kindly adjust as follow: “shows”.

Response: We have corrected “showed” into “shows” in our revised manuscript. Thank you.

5) 4.2. Salt stress: Page 8, line 289: Kindly adjust as follow: “exhibit”.

Response: We have amended this problem in our revised manuscript. Thank you.

6) 4.3. Nutrient stress: Page 8, line 294: Kindly adjust as follow: “phosphorus (P)”. Kindly pay attention to adjust this symbol in the whole manuscript.

Response: Thank you for your kindly suggestion. We have adjusted “phosphorus (Pi)” to “phosphorus (P)” in our manuscript. And we have checked this issues in the full text to avoid the same problem. Thank you again.

7) 4.3. Nutrient stress: Page 8, line 298: Kindly adjust as follow: “has been reported”.

Response: We have corrected this problem in our revised manuscript. Thank you.

8) 4.3. Nutrient stress: Page 8, line 314: Kindly adjust as follow: “enhanced”.

Response: We have corrected this mistake in our manuscript. Thank you.

9) 4.3. Nutrient stress: Page 8, line 322: Kindly adjust as follow: “controlled”.

Response: We have corrected this problem in our manuscript. Thank you.

10)  4.4. Light and gravity: Page 9, line 342: Kindly adjust as follow: “regulated”.

Response: We have corrected this mistake in our manuscript. Thank you.

  1. The regulatory roles of MYB in roots are mediated by various phytohormones

 1)  Pages 9–10, lines 345–392: “The regulatory roles of MYB in roots are mediated by various phytohormones” part is well structured and aiming.

Response: Thank you very much for your positive comments about this part.

  1. Conclusions and Prospects

 1)  Pages 10–11, lines 394–421: The Conclusions and Prospects part is very well structured and aiming. However, I suggest removing all in-text references found in this part as this is not appropriate.

Response: According to your requirement, we have already removed all of the references in this part in our manuscript. Thank you.

2) Page 10, line 409: Kindly remove “And” from the beginning of the sentence.

Response: We have already removed it in our revised manuscript. Thank you very much again.

Reviewer 2 Report

This review manuscript covers a good narrative about MYB transcription factors, pinpointed specifically to plant roots. It is indeed, there are so many previous studies on this gene that covers many aspects from development, stress responses and hormonal responses that have been covered. In my opinion, I think it will be more interesting if the authors add more details about this TF from previous studies that are also more specific for the root. For instance, in the Introduction section, the authors explain why did they focused solely on the root at the first paragraph that the root system is plant's greatest interface to its environment, the soil environment that is highly complex as there are biotic and abiotic factors. But I could not find anything about biotic factors, such as stresses by pests (e.g., nematodes, insects, etc.,) and diseases (e.g., virus, bacteria, etc.). I would like to see if there any study done about it and screened for MYB using qRT-PCR. If not, I strongly suggest at the end of the section, just before the conclusion, the authors added extra information about future studies. I'd like to suggest some details to be added, as I think there are some missing information and to further, fortify the manuscript itself.

Introduction

L34-36: Don't forget, biotic and abiotic stress factors too

L41-54: I suggest the authors to start with the general information about MYB first instead placing the fact about MYB being existed in animal cell too (L53-54). Note that MYB also exists across the realm of viruses, animals, and plants. Define the MYB, the structure, the overall function of each types. Refer specifically what are they role in virus (1-2 short explanatory sentences), animals , which species and the functional examples (1-2 sentences), before you move to plants. After this, you can fluidly talk about the MYB in plants.

Section 2-3

The authors made a good explanations about the role of MYB gene on Arabidopsis. However, is that it? No example from another plants? I think they already have studied anything on root development on rice, too (please see: https://doi.org/10.1093/jxb/erx174). Please add more examples from other plant subjects to strengthen your narrative. Even there are still minimal studies, the authors can still add/refer it and this will be consistent to the Conclusion and Prospects section where the authors mentioned that many studies are still focused on Arabidopsis and more plant species require further analysis in the future.

Section 4

Be careful, for scientific format reason, there are many plant genera and species in this section are not italicized

Subsection 4.4

L337-339: Could you elaborate more about the gravitational stimulation here? For example, was the experiment done in ISS (in situ, in space) or ground-based simulation (e.g., using clinostat)? Was it microgravitational simulation or higher-G? If so, how many G-force?

Note between section 4 to 5: I strongly suggest for the authors to add another section here about biotic stress experienced by the root. I found so many studies have reported their cases about the MYB and the responses during nematode attack, fungi, pests (insects), or even plant viruses. If any of the case I referred here are still have minimal studies reported (or even none), write it down as potential future studies suggested to the readers. Perhaps, at the end of each section, anything that are still minimally studied, the authors can refer it before it will be wrapped up at the very end of the Conclusion and Prospects section.

Author Response

Dear Reviewer#2,

Thank you very much for reviewing our manuscript as well as for your kindly suggestions in our manuscript “MYB transcription factors becoming mainstream in plant roots” (Manuscript ID: ijms-1860833). The comments of you were highly insightful and enabled us to improve the quality of our manuscript. We have tried our best to address all of your requirements, carefully. In the following pages are our point-by-point responses to each comment of you. According to your suggestion, we have added some new contents about the functions of MYB genes involved in roots in none-model plants, and have also added a new part about the functions of MYB genes in response to biotic stress in roots in the revised manuscript (Page 10). As a result, the number of references in the main text and the “Reference” parts were adjusted accordingly. Please check them. Thank you.

According to the journal’s requirement, all of the revisions in the manuscript were marked up using the “Track Changes” function in the Microsoft word. Moreover, we use red highlight for the main amends in the manuscript. We hope that our revisions in the manuscript and our accompanying responses will be sufficient to make our manuscript suitable for publication in your journal.

Thank you very much for your hard work in reviewing our manuscript.

Yours sincerely,

Hai Du

Reviewer#2

Comments and Suggestions for Authors:

This review manuscript covers a good narrative about MYB transcription factors, pinpointed specifically to plant roots. It is indeed, there are so many previous studies on this gene that covers many aspects from development, stress responses and hormonal responses that have been covered. In my opinion, I think it will be more interesting if the authors add more details about this TF from previous studies that are also more specific for the root. For instance, in the Introduction section, the authors explain why did they focused solely on the root at the first paragraph that the root system is plant's greatest interface to its environment, the soil environment that is highly complex as there are biotic and abiotic factors. But I could not find anything about biotic factors, such as stresses by pests (e.g., nematodes, insects, etc.,) and diseases (e.g., virus, bacteria, etc.). I would like to see if there any study done about it and screened for MYB using qRT-PCR. If not, I strongly suggest at the end of the section, just before the conclusion, the authors added extra information about future studies. I'd like to suggest some details to be added, as I think there are some missing information and to further, fortify the manuscript itself.

Introduction

L34-36: Don't forget, biotic and abiotic stress factors too

Response: Thank you for your suggestion. As will be mentioned below, we have added a ne part about the functions of MYB genes in response to biotic stress in roots in the revised manuscript (Page 10), and we have changed “abiotic stress” to “biotic and abiotic stresses” in the “Keywords” and “Introduction” parts. Thank you, again.

L41-54: I suggest the authors to start with the general information about MYB first instead placing the fact about MYB being existed in animal cell too (L53-54). Note that MYB also exists across the realm of viruses, animals, and plants. Define the MYB, the structure, the overall function of each types. Refer specifically what are they role in virus (1-2 short explanatory sentences), animals, which species and the functional examples (1-2 sentences), before you move to plants. After this, you can fluidly talk about the MYB in plants.

Response: Thank you for your suggestion. We have added the general information of MYB transcription factor genes, and have added the suggested contents about their functions in none-plant species in this part. Thank you.

Section 2-3

The authors made a good explanations about the role of MYB gene on Arabidopsis. However, is that it? No example from another plants? I think they already have studied anything on root development on rice, too (please see: https://doi.org/10.1093/jxb/erx174). Please add more examples from other plant subjects to strengthen your narrative. Even there are still minimal studies, the authors can still add/refer it and this will be consistent to the Conclusion and Prospects section where the authors mentioned that many studies are still focused on Arabidopsis and more plant species require further analysis in the future.

Response: Thank you for your great suggestion. Yes, in addition to model plant Arabidopsis, many genes in other plant species have been functionally characterized to date, especially in rice. However, as mentioned in the “Introduction” part, up to now, the majority of the known functions of MYB genes center on their roles in controlling the transcriptional networks of a number of biological processes in the aboveground parts, such as plant development, secondary metabolism, and biotic and abiotic stress processes, while a relative less genes functions in roots directly. Given the large number of this super gene family in plant genome and an increasing number of their characterized functions in roots, it is foreseeable that there are a much larger number of MYB genes function in distinct biological processes in roots. Thus, in this review, we summarized the current knowledge about the functions of MYB genes in roots.

According to your suggestion, we have carefully checked the relevant literatures in the past decades in PubMed in NCBI, and have added the research advance of MYB genes related to roots in the revised manuscript, such as line 145-151 in page 4, line 227-233 in page 5, line 227-233 in page 5. Meanwhile, we have checked the reference that you suggested (https://doi.org/10.1093/jxb/erx174), and we found that the OsMYB1 gene in this paper is involved in root development under Pi starvation. And we have cited the paper in the section of “ 4.3. Nutrient stress”. Thank you for your kindly suggestion again.

Section 4

Be careful, for scientific format reason, there are many plant genera and species in this section are not italicized

Response: We are sorry for this mistake. We have checked this issue in our manuscript carefully, and have corrected it in the revised manuscript if it was necessary. Thank you.

Subsection 4.4

L337-339: Could you elaborate more about the gravitational stimulation here? For example, was the experiment done in ISS (in situ, in space) or ground-based simulation (e.g., using clinostat)? Was it microgravitational simulation or higher-G? If so, how many G-force?

Response: Thank you for your comment. In the paper, the experiment that you mentioned was performed by rotating vertically grown 4-day-old seedlings by 90° on MS medium to measure the growth responses of primary root to gravity stimulation, that is, gravity stimulation means reorientation of 90°. According to your comment, we have mentioned the methods of this study in the revised manuscript (line 371-377, page 9). Thank you.

Note between section 4 to 5: I strongly suggest for the authors to add another section here about biotic stress experienced by the root. I found so many studies have reported their cases about the MYB and the responses during nematode attack, fungi, pests (insects), or even plant viruses. If any of the case I referred here are still have minimal studies reported (or even none), write it down as potential future studies suggested to the readers. Perhaps, at the end of each section, anything that are still minimally studied, the authors can refer it before it will be wrapped up at the very end of the Conclusion and Prospects section.

Response: Thank you for your insightful suggestion. It is good idea. Yes, MYB genes are widely involved in biotic stress processes, but most of their known functions are in aboveground parts rather than roots. According to your requirement, we have added a new part (“4.5. Biotic stress” in page 10) about the functions of MYB genes in response to biotic stress in roots in the revised manuscript. As a result, the number of references in the main text and the “Reference” parts were adjusted accordingly. Please check them. Thank you again.

Round 2

Reviewer 2 Report

In my perspective, the authors have done a good job to revise the manuscript. The manuscript now, I think, is comprehensive yet not abundant. Have a good flow to read. With more plant examples and added details to each factors, I believe this manuscript can be a good model for both younger and experienced researcher to learn from it or to update their knowledge.